# Sliding Dynamics of Ring Chains on Two Asymmetric/Symmetric Chains in a Simple Slide-Ring Gel

**DOI:** 10.3390/polym14010079

**Published:** 2021-12-26

**Authors:** Jiaxin Wu, Fuchen Guo, Ke Li, Linxi Zhang

**Affiliations:** Department of Physics, Zhejiang University, Hangzhou 310027, China; 21936020@zju.edu.cn (J.W.); 11836024@zju.edu.cn (F.G.); 21736027@zju.edu.cn (K.L.)

**Keywords:** sliding dynamics, molecular dynamics simulation, diffusion coefficient, sub-diffusion, slide-ring gel

## Abstract

The sliding dynamics along two asymmetric/symmetric axial chains of ring chains linked by a linear chainis investigated using molecular dynamics (MD) simulations. A novel sub-diffusion behavior is observed for ring chains sliding along eithera fixed rod-like chain or fluctuating axial chain on asymmetric/symmetric axial chainsat the intermediate time range due to their strongly interplay between two ring chains. However, two ring chains slide in the normal diffusion at along time range because their sliding dynamics can be regarded as an overall motion of two ring chains. For ring chains sliding on two symmetric/asymmetricaxial chains, the diffusion coefficient D of ring chains relies on the bending energy of axial chains (K_b_) as well as the distance of two axial chains (d). There exists a maximum diffusion coefficient D^max^ at d = d* in which ring chains slide at the fastest velocity due to the maximum conformational entropy for the linking chain between two ring chainsat d = d*. Ring chain slide on fixed rod-like axial chainsfaster in the symmetric axial chain case than that in the asymmetric axial chain case. However, ring chains slide on fluctuatingaxial chainsslower in the symmetric axial chain case than that in the asymmetric axial chain case. This investigation can provide insights into the effects of the linked chain conformation on the sliding dynamics of ring chains in a slide-ring gel.

## 1. Introduction

Rotaxanes, firstly reported by Harrison et al. [1], are a class of mechanically interlocked molecules (MIMs) that are composed of ring chains and axial polymer chains bound together in a threaded structure.These moleculeshave attracted much attention for the potential advantages of the dynamic nature of the mechanical bond as a consequence of the low energy barrier for the rotation of the ring chains, or the sliding of the ring chains along the axial polymer chain [2]. Owing to their fascinating sliding dynamics, rotaxanes can be applied to molecular machines. For example, molecular shuttles have been developed according to the fact that the ring chain can move from onerecognition site to another along the axial chain in response to external stimuli such as pH [3] temperature, [4] and light [5]. With the mechanical penetration of the axial chain into the ring chains, rotaxanes exhibit a fascinating mobility where ring chains can slide freely on the axial chain, the position of ring chain can be controlled well by the energy potential along the axial chain corresponding to the interaction between the ring chain and the axial chain [6,7]. The sliding motion of ring chains on the axial chain enables us to fabricate functional rotaxane-based devices and materials [8,9,10,11].

As we know, a notable feature of mechanically interlocked polymers (MIPs) such as polycatenanes and polyrotaxanes is the relative sliding motion between the mechanically bonded components. Therefore, exploring the sliding dynamics of the ring chain in polycatenanes or polyrotaxanes is essential to fundementally understandthe properties and structures of MIPs. Zhang et al. investigated the dynamics behaviors of ring chains by coarse-grained molecular dynamics, and found that the sliding-ring system has much lower mobility incomparison with the traditionally fixed junction system [12]. Yasuda et al. reported the sliding dynamics of a ring chain along an axial chain in rotaxanes with different ring sizes and axial chain conformations and revealed that the sliding dynamics of the ring chain relies not only on the friction between the ring chain and the axial polymer chain but also on the local fluctuation of the axial chain [13]. Kato et al. investigated the effects of ring size on the dynamics of polyrotaxane glass using viscoelastic measurements and pointed out thatthe dynamics of glass transition in polyrotaxanes can be controlled by the size of the ring chains, and not only free volume such as the interchain interactions or the interchain gap but also the intrachain gap located between different components affect the transition dynamics of polyrotaxane glass [14].Rauscher et al. used a “double-Rouse” model to simulate the dynamics of unentangled poly[n]catenanes and obtainedthe motion characteristics of the monomer in each time period with a Rouse-like motion mode, providinga prediction of the correct scaling behavior of the ring and chain relaxation times [15]. Wang et al. successfully reported the dynamics of the ring chain threaded on the axial chain through a mesoscopic simulation method and found that the hydrodynamic interactions significantly enhance the diffusion and relaxation of the threading ring chains [16]. We also investigated the sliding dynamics of multi-rings on a semiflexible chain in catenanes using molecular dynamics simulation and explored the influences of the bending energy of axial polymer chain and the number of threading ring chains on the sliding process of ring chains [17]. We performed further investigations to study the sliding dynamics of ring chains on a knotted polymer chain in rotaxane and found that the sliding behavior of the ring chain exhibits a sub-diffusion at the intermediate time range resulting from the intra-strand interaction between the ring chain and the knotted axial chain in the knot crossing region [18].

Recently, Okumura et al. fabricated ‘slide-ring (SR) gels’ by cross-linking the α–cyclodextrins of polyrotaxanes in a solution in which the polymer chains are cross-linked by figure-of-eight-shaped molecules consisting of two ring chains [19]. These cross-linking junctions are not fixed and can move freely in the polymer network. The fluctuation of the cross-links can reduce the Young’s moduli of the SR gels and results in high softness compared with conventional chemically cross-linked gels [20,21]. For usual fixed cross-links (FC) gels, with increasing cross-linking density, the Young’s modulus increases and fracture energy decreases. However, SR gels show unique independence of fracture energy regardingthe Young’s modulus [22]. In addition, Kato et al. found that the mechanical relaxation is attributable to the sliding dynamics of the chains through cross-links [23]. The sliding of the cross-linking point along the chain at a crack tip in SR gels can enlarge strand length between the cross-links and increase the fracture toughness of the SR gels [23].

The sliding dynamics of ring chains in mechanically interlocked polymers is crucial for understanding the origin of both the softness and toughness of SR gels. Some previous simulation works haveonly concentrated on the sliding dynamics of isolated ring chains along the axial chain in which two ring chains arenot linked by a polymer chain [12,13,14,15,16,17,18]. In fact, two ring chains are always linked by a polymer chain in SR gels [19,20,21,22,23], and the sliding dynamics of one ring chain is closely related tothe sliding motion of another linked ring chain. Therefore, it is important to explore the sliding dynamics of two linked ring chains along axial chains, determining the interior relationship between the sliding dynamics of these two linked ring chains and the conformations of the linking chain between two ring chains, as well as the properties of two axial chains. Our aim is to understand the effects of the interplay between two ring chains on the sliding dynamics of ring chains along the axial chain. Our simulation results show that the linked structure of two ring chains seriously affects the sliding dynamics of ring chains along the axial chain.

## 2. Model

### 2.1. Coarse-Grained Model of Rotaxane

In our simulation, a bead-spring chain developed by Kremer and Grest is used to model a simple side-ring gel [24]. The chain consists of two rigid ring chains and two axial polymer chains, in which rigid ring chains are threaded by the axial polymer chain.In addition, one rigid ring chain is linked with another by a linear polymer chain, see Figure 1. Moreover, a real snapshot of the simulation system using Visual Molecular Dynamics (VMD) is also shown in Figure 1B [25]. The number of monomers of the rigid ring chain and axial chain is N_ring_ and N_axial_, and the number of monomers of the linking chain between two ring chains is N_linking_. In our simulation, N_ring_, N_axial_ and N_linking_ are fixed as N_ring_ = 12, and N_axial_ = N_linking_ = 100. The results for N_axial_ = 200 are givenin the Appendix A. In order to prevent the overlap between monomers, a shift and cut-off Lennard–Jones (LJ) potential is adopted for any two monomers [26].
(1)$$U_{\mathrm{LJ}}(r)=\{\begin{array}{cc} 4\varepsilon \left[{\left(\frac{\sigma }{r}\right)}^{12}-{\left(\frac{\sigma }{r}\right)}^{6}+\frac{1}{4}\right] & r<r_{C}\\ 0 & r>r_{C} \end{array}$$
where the cut-off distance r_C_ is chosen to be 2^1/6^σ, and ε is set as 1 k_B_T. Here σ is the diameter of monomers, k_B_ is the Boltzmann constant and T is the temperature.

All neighbouring monomers in all chains are connected by the finitely extensible nonlinear elastic (FENE) potential [27,28,29].
(2)$$U_{\mathrm{FENE}}(r)=-\frac{{\mathrm{KR}}_{0}^{2}}{2}\ln\left[1-{\left(\frac{r}{R_{0}}\right)}^{2}\right]$$
where the spring constant K is chosen to be 30 k_B_T/σ^2^ and the maximum extension of the spring R_0_ is R_0_ = 1.5σ.

The chain stiffness is introduced by means of a bending potential between two adjacent bonds [13,17].
(3)$$U_{\mathrm{bending}}(\theta )=\frac{1}{2}K_{\mathrm{bending}}(\theta -\theta_{0}{)}^{2}$$
where θ is the angle between two consecutive bonds, and K_bending_ is the bending constant, which represents the bending energy. Here the θ_0_ is fixed to be θ_0_ = π[(N_ring_ − 2)/N_ring_] for ring chains, and θ_0_ = π for the two axial chains and the linking chain [13,14,15,16,17,30,31]. The bending energy K_bending_ is chosen to be 100 K_B_T/rad^2^ for ring chains. In our model, ring chain A is threaded on the left axial chain with the bending energy of K_1_ and ring chain B is threaded on the right axial chain with the bending energy of K_2_ while these two ring chains are linked by a flexible linear chain with the chain length of N_linking_ as well as the bending energy of K_linking_ = 0. The distance of the two axial chains is d, which is varied from d = 2 to d = 60. A schematic diagram of two ring chains sliding along two axial chains in a simple slide-ring gel is shown in Figure 1. The chain ends are connected under periodic boundary conditions, and each axial chain is a long periodical chain.

### 2.2. Molecular Dynamics Simulation

We performed molecular dynamics (MD) simulations by using the large-scale atomic/molecular massively parallel simulator (LAMMPS) [32]. Reduced units of ε = 1, m = 1, σ = 1 and τ_0_ = (mσ^2^/k_B_T)^1/2^ = 1 are chosen to be the units of energy, mass, length and time, respectively. The timestepτ is chosen to be τ = 0.005τ_0_, and the constant temperature is fixed as T_0_ = 1.0 in the unit ofε/k_B_ controlled by the Langevinthermostat. The motion of all monomers is described by means of the following Langevin equation [13,17,18,33].
(4)$$m\frac{d^{2}r_{i}}{{\mathrm{dt}}^{2}}=F_{i}-\;m\gamma \frac{{\mathrm{dr}}_{i}}{\mathrm{dt}}+W_{i}\left(t\right)$$
where r_i_ is the coordinate of the i-th monomer, and the friction constant is set as γ = 0.5τ_0_^−1^. F_i_ is the resultant force acting on the i-th monomer from all other elements, including the Lennard–Jones, FENE and the bending potentials described above. The quantity W_i_(t) is the random force applied to the i-th monomer with a zero mean and satisfies the following fluctuation:<W_i_(t)W_j_(t′)>=2 k_B_Tmδ_ij_δ(t−t′)(5)

In order to characterize the sliding dynamics of the ring chain, we calculated the mean-squared distance (MSD) of the center-of-mass of the ring chain, g_3_(t), in the coordinate index along the axial chain contour path for a time range up to ~10^6^ [13,17,18]. Here the coordinate index is recorded by finding the closet monomer of the axial chain to the ring chain from one end to the other, and the g_3_(t) is defined as [13,17,18]:g_3_(t) = σ^2^<i(t)−i(0)>^2^(6)

Here i(t) represents the index number of the monomer of the axial chain, which is closest to the ring chain at a time t, and σ is the bond length. In fact, the unit of g_3_(t) is σ^2^. Therefore, the diffusion coefficient D is estimated from the g_3_(t) as [13,17,18,34]:(7)$$D=\underset{t\to \infty }{\lim}\frac{g_{3}\left(t\right)}{2\;t}$$

All data for g_3_(t) are averaged by conducting 200 simulation runs, which is assumed to providegood statistics.

## 3. Results and Discussion

### 3.1. Sliding Dynamics of Ring Chains on Two Symmetric Chains (K_1_ = K_2_)

We first investigated the sliding dynamics of ring chains on the symmetric fixedrod-like chains (i.e., K_1_ = K_2_ = ∞), and the results are given in Figure 2 and Figure 3. Figure 2 showthat the subdiffusion behavior occurs at the intermediate time range of 100 < t < 3000. As the subdiffusion behavior reflects the interplay of two ring chains during the sliding process, the almost same subdiffusion region for different distances d between two symmetric fixedrod-like chains indicates the interplay of two ring chains occurs after the ring chain has slid the same distance. At the longer time range (t > 30,000), as two ring chains diffuse in an overall motion and the interplay between two ring chains havedisappeared during their sliding process, a long-time normal diffusion is shown. Normal diffusion occurs for ring chains on two fixed rod-like chains at a longer time range, and a non-monotonic increase for g_3_(t) is shown in the inset figure in Figure 3 when d increases from d = 2 to 60. In Figure 3, there exists a maximum value of D^max^ located at d* = 16. In this case, the ring chain slides at the fastest speed along the axial chain because the effect of the linking flexible linear chain between two ring chains for d = d* is minimized. When d increases further, the linking flexible linear chains between two ring chains haveto be stretched, the resistance for the sliding process of one ring chain from another ring chain strengthens again and the diffusion coefficient D decreases.

We also discussthe sliding dynamicsof two linked ring chains on the symmetric fluctuating chains(K_1_ = K_2_ = 0).In this case, the distance of two fixed points (such as AB or CD in Figure 1) for the fluctuating axial chain is 19σ, the average end-to-end distance of a single flexible linear chain with the chain length of 100 monomers in a dilute solution [13,17].To characterize the sliding dynamics of ring chains in a simple slide-ring gel, we calculatedthe g_3_(t) of ring chain A, and the result is shown in Figure 4. As ring chains A and B are symmetric in this case, the sliding diffusion behaviors of ring chains Aand B are also the same, and only the g_3_(t) of ring chain A is given in Figure 4. At a shorter time range of t < 200, the g_3_(t) is proportional to t, which means that the interplay between two ring chains can be ignored, and the sliding diffusion of two ring chains along flexible axial chains is normal. However, at the intermediate time range (6000 < t < 50,000), for 40 > d > 5, the g_3_(t) exhibits a sub-diffusion behavior with g_3_(t)~t^α^ (the value of αis less than 1) owing to the fact that two ring chains are linked by a flexible linear chain and these two linked ring chains interact on each other at this intermediate time range. At the longer time range (t > 50,000), the interplay between two ring chains in the diffusion has disappeared, and ring chains diffuse in an overall motion of two ring chains, showing a long-time normal diffusion. For d = 2, as two axial flexible chains are next to each other, the effects of neighboring axial flexible chains on the sliding dynamics of ring chains are considerable, which leads to a very long sub-diffusion region for the sliding dynamics of ring chains on axial chains. Meanwhile, for d = 60, as two axial flexible chains are far enough and the linking chain between two ring chains has to be stretched during their sliding process, the sub-diffusion occurs at t = 500, which is much earlier than t = 6000 for d = 5, and the ring chain slides more slowly than that for d = 5~40.

The sliding dynamics could be discussed in terms of diffusion coefficient D, and Figure 5 show the dependence of D on the distance d between two axial chains. When d increases from d = 2 to 16, D increases from D = 0.015 to 0.075, and there exists a maximum value for D^max^ located at d* = 16. Compared with Figure 3, we can find that the same value of d* for ring chains sliding on either the symmetric fluctuating chain or the symmetric fixedrod-like chains indicates that the value of d = d* = 16 only relies on the chain length of the linking chains between two ring chains.When d increases from d = 2 to d = 16, D increases monotonically, and D decreases monotonically from d = 16 to d = 60. Comparing with Figure 3, Figure 5 show that the diffusion coefficient D of ring chain on the symmetric fixedrod-like chains is much less than that on the symmetric fluctuating chains [13,17], for example, D^max^ = 0.017 for ring chains on the symmetric fixedrod-like chains, which is only 23% of D^max^ = 0.075 for ring chains on the symmetric fluctuating chains. This result is in good agreement withthe previous result of the sliding dynamics of ring chains on a polymer chain in rotaxane [13,17]. The ring chain on the fixed rod-like chain diffuses freely in one dimension, whereas the ringchain on the fluctuating axial chain diffuses in quasi-three dimensions, which leads to slidingalong the fluctuating axial chain faster than that along the fixed rod-like chain. These results indicate that the sliding dynamics aredominated not only by the diffusion of the ring chain but also by the local fluctuating motion of the axial chain as well as by the interplay of two ring chains through the linking chain between two ring chains.

In order to explain the reason why the ring chain slides fastest along the axial chain at d = d*, we calculated the probability distribution functionP(r) of end-to-end distance for a single 100-monomer flexible polymer chain, and the results are shown in Figure 6a. The probability distribution function P(r) is directly related to the Helmholtz free energy F(r) of a chain having the end-to-end distance r by [35,36,37,38]:F(r) = C−k_B_TlnP(r)(8)
where C is a constant. From Figure 6a, we can find that the maximum value of P(r) is located at r* = 16.5, which is very close to the position of the peak for the diffusion coefficient D at d = d* = 16 in Figure 3 and Figure 5. As the Helmholtz free energy reaches the minimum value at r = r*, the entropy force acting on the ring chains from the linking flexible chain approaches to the minimum value at r = r*. If the distance between two axial chains d is less than r*, or greater than r*, there exists an extra entropy force originated from the conformational entropy of the linking flexible chain between two ring chains to act on the ring chains, and this extra entropy force can strengthen the friction force between ring chains and the axial chain. Of course, this extra friction force is close to zero at d = d* owing to the fact that the conformational entropy of the linking chain between two ring chains approaches the maximum, which leads to the fastest sliding diffusion at d = d* for ring chains on the axial chain. A simple explanation is given in Figure 6b.

### 3.2. Sliding Dynamics of Ring Chains on Two Asymmetric Chains (K_1_ ≠ K_2_)

In this section, we focus on the sliding dynamics of two rings on the asymmetric axial chains, and we set the left axial chain I in Figure 1 as a fixed rod-like chain with K_1_ = ∞ and the right axial chain IIas a fluctuating one with k_2_ = 0. In this case, the distance between two fixed points of C and D is also fixed to be 19σ. We calculated the g_3_(t) of two ring chains sliding along the asymmetric axial chains, and the results are shown in Figure 7. In Figure 7a, it is also found that at the intermediate time range of 1000 < t < 30,000, ring chain sliding along the fixed rod-like axial chain on the asymmetric axial chains also displays the subdiffusion behavior with the scaling relationship of g_3_^I^(t)~t^0.53^, which is completely different from the sliding dynamics behavior of a single isolated ring chain on a fixed rod-like chain [13]. However, it is almost the normal diffusion for ring chain sliding along the fluctuating chain on the asymmetric axial chain case for d = 5, 16, and 30 due to the fact that the ring chain along the fluctuating chain slides faster than that along the fixed rod-like chain on the asymmetric axial chains and the interplay between two ring chains nearly disappears for d = 5, 16, 30, see Figure 7b. Moreover, it also shows a subdiffusion behavior for ring chains along the fluctuating chain on the asymmetric axial chains for d = 2 or d = 40 because the interplay between two ring chains always exits for d = 2 or d = 40. At the shorter time range, ring chains slide in the normal diffusion because the interplay between two ring chains isnot yet reflected at this time. At the longer time range, such as t > 100,000, two ring chains on the asymmetric axial chains show the normal diffusion of $g_{3}^{A}(t$) or $g_{3}^{B}\left(t\right)$ ~ t^α^ with α = 1.0 owing to the fact that two ring chains diffuse in an overall motion, and the interplay between two ring chains has disappeared at t > 100,000. Meanwhile, a non-monotonic dependence of D on the distance d between two axial chains is shown in Figure 8a, which is the same as the sliding dynamics of ring chains on the symmetric chains, see Figure 3 and Figure 5. The same critical value of d* = 16 for the diffusion coefficient D of ring chains on the asymmetric axial chains confirms again that the conformational entropy of the linking flexible chain between two ring chains seriously affects the sliding dynamics of ring chains on the asymmetric axial chains. The ratio of D^B^/D^A^ for chain Bsliding along the fluctuating chain and ring chain Asliding alongthe fixed rod-like chainon theasymmetric axial chain case is shown in the inset figure in Figure 8a; and we found that the ratio of D^B^/D^A^ keeps a constant of 0.5 for different d. In fact, the overall motion of two ring chains needs the same speed in the *z*-axis direction. Although the diffusion coefficient D is measured along the contour length of the axial chains, the 100 monomers of the flexible axial chain span a vertical distance (i.e., the *z*-axis distance) equal to 19σ. We can obtain the theoretical value of D^B^/D^A^ equals approximately to D^B^/D^A^~(19/99)^2^ = 0.037, and this theoretical value is independent of the distance d. Our simulation results are in agreement with this theoretical value.

In order to quantitatively compare the sliding velocities of ring chains on the asymmetric axial chains and on the symmetric axial chain, we calculate the ratio of D^asymmetric^/D^symmetric^ for ring chains along the fixed rod-like axial chain or fluctuating axial chain on asymmetric/symmetric axial chains, and the results are shown in Figure 8b. For fixed rod-like axial chains, D^asymmetric^/D^symmetric^ is always less than 0.5, which means that the ring chain slides along the fixed rod-like axial chain slower on asymmetric axial chains than on symmetric axial chains. In order to explain the reason why the diffusion coefficient D of ring chain B along fixed rod-like chain on asymmetric axial chains is less than that on symmetric axial chains, we calculated the absolute diffusion of the ring along the fluctuating axialchains in the *z*-axis direction by the mean-square displacement (MSD), $g_{3}^{z}\left(t\right)$, of the ring from the center of mass of the ring in the *z*-axis direction.
(9)$$g_{3}^{z}\left(t\right)={<[z\left(t\right)-\;z\left(0\right)]}^{2}>$$
where z(t) is the z-coordinate value of the center of mass of the ring chain at time t. For comparison, we also calculated the absolute diffusion of ring chains along the symmetric fixed rod-like axial chains in the *z*-axis direction. Figure 9a show that although the sliding dynamics of ring chains along the symmetric fluctuating axial chains is faster than that along the symmetric fixed rod-like axial chains (see Figure 3 and Figure 5), the absolute diffusion of the ring along the symmetric fluctuating axial chains in the *z*-axis direction is slower than that along the symmetric fixed rod-like axial chain. The reason is that the sliding diffusion coefficient is measured along the contourlength of the axial chains. Thus, while the 100 beads of the flexible chainspan a verticaldistance equal to 19σ, only 19 beads of the rigid chainare necessary to span the same distance. According to the results of Figure 9a, we can determinethat given a time period ∆t, it is ∆Z_3_ > ∆Z_1_ for the ring chain along the symmetric rod-like chains and along the symmetric fluctuating chains, where ∆Z_1_ is the projection distance along *z*-axis direction, and the *z*-axis direction is parallel to the fixed rod-like direction, see Figure 9b(I–III). Once two ring chains slide on the asymmetric axial chains, see Figure 9b(II). As two chains slide in the overall motion, the synchronous motion leads to a relative relationship of ∆Z_3_ > ∆Z_2_ > ∆Z_1_. We can make a conclusion that there exists a relative relationship of S′_1_ > S_1_, and S′_3_ < S_3_, here S′_1_, S_1_, S′_3_ and S_3_ are their contour distances, see Figure 9b. Therefore, this schematic illustration can explain clearly the reason why D^asymmetric^/D^symmetric^ is greater than 1 for ring chain sliding on the fluctuating chain and D^asymmetric^/D^symmetric^ is less than 1 for ring chain sliding on the fixed rod-like chain, see Figure 8b.

## 4. Conclusions

The sliding dynamics of ring chains along rod-like and fluctuating axial chains on the asymmetric/symmetric chains in a simple slide-ring gelis studied by usingthe MD simulation method. The mean-square displacement (MSD) of the center-of-mass g_3_(t) of the ring chain and the diffusion coefficient D of the ring chain can be used to characterize their sliding dynamics. A novel subdiffusion behavior is observed for ring chains sliding along either fixed rod-like or fluctuating axial chains on the asymmetric/symmetric chains at the intermediate time range, indicating that the interplay between two ring chains through the linking flexible chain affects the sliding dynamics seriously during the intermediate time range. When ring chains slide along two symmetric axial chains, the diffusion coefficient D of the ring chain relies on the bending energy of axial chains (K_b_) and the distance d of two axial chains. There exists a critical distance d* at whichthe ring chain slides fastest with a maximum diffusion coefficient D^max^ owing to the fact that the linking chain between two ring chains hasthe maximum conformational entropy at d = d*, and the friction force between ring chain and the axial chain approaches to the minimum value due to lacking the extra entropy force. When ring chains slide along two asymmetric axial chains, the subdiffusion behavior strengthens for the ring chain along the fixed rod-like axial chain and weakens for the ring chain along the fluctuating axial chain. Meanwhile, it shows a normal diffusion for ring chain along either fixed rod-like axial chain or fluctuating axial chain at the longer time range. The ring chain slides along the fluctuating axial chain on the asymmetric chains faster than that on the symmetric chains;however, ring chains diffuse along the fixed rod-like axial chain on the asymmetric chains slower than that on the symmetric chains. Moreover, the non-monotonic dependence of diffusion coefficient D on the distance d between two axial chains for ring chains along asymmetric axial chains also confirms that the conformational entropy of the linking flexible chain between two ring chains seriously affects the sliding dynamics of the ring chain. Our results can help us understand the complex sliding behavior for ring chains in a slide-ring gel.

## Figures and Tables

**Figure 1 polymers-14-00079-f001:**
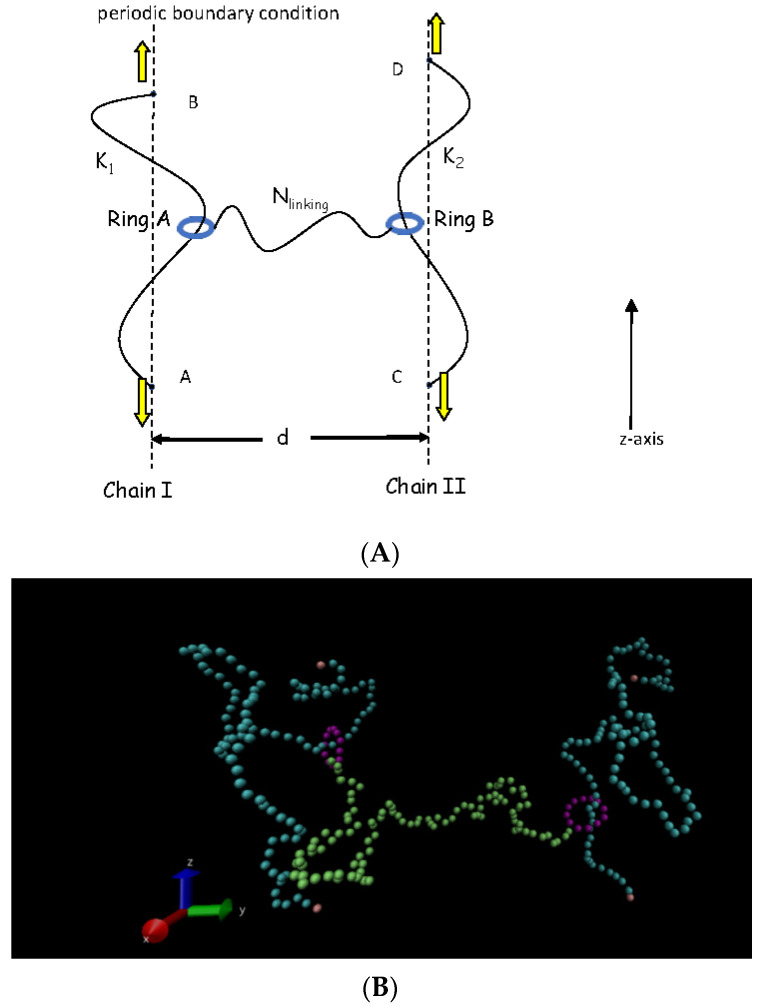
(**A**) Schematic diagram of a simple slide-ring gel in which two ring chains are linked by a flexible linear chain with the chain length N_linking_ and two ring chains slide along two axial chains with different bending energies of K_1_ and K_2_ as well as the distance of two parallel axial chains is d. Points A, B, C and D are fixed, and the line AB is parallel to the line CD [13,18].(**B**)A real snapshot of simulation system by using visual molecular dynamics (VMD) for ring chains sliding along the symmetric fluctuating axial chains [25].

**Figure 2 polymers-14-00079-f002:**
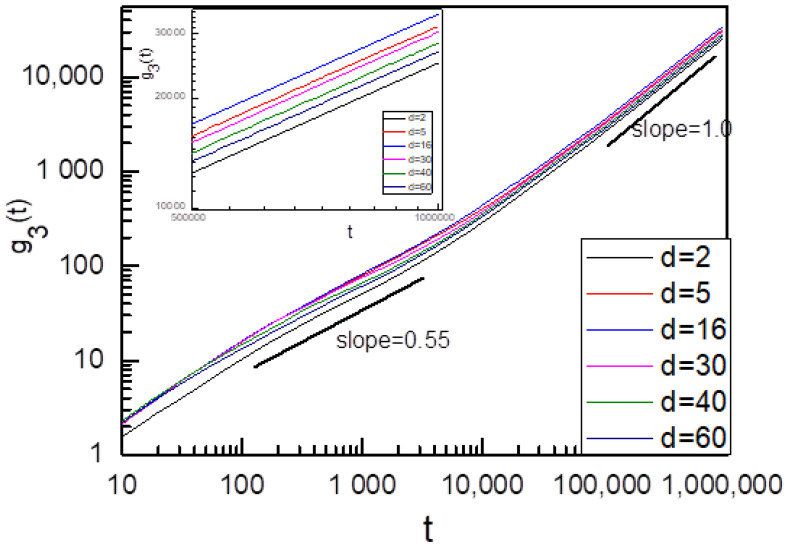
Time dependence of g_3_(t) of ring chain on the symmetric fixed rod-like chains (K_1_ = K_2_ = ∞) with different distances d between two symmetric fixed rod-like chains.

**Figure 3 polymers-14-00079-f003:**
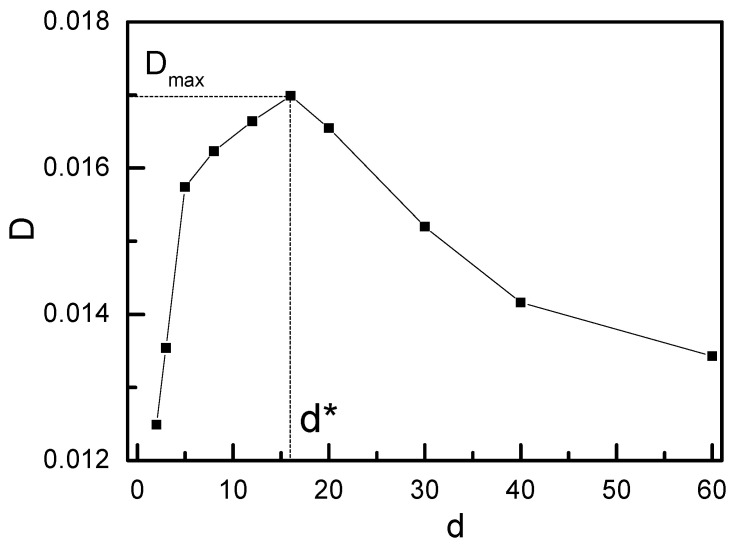
Diffusion coefficient Dof ring chain on the symmetricfixed rod-like chains(K_1_
*=* K_2_
*=* ∞) with different distances d between two symmetricfixed rod-like chains.

**Figure 4 polymers-14-00079-f004:**
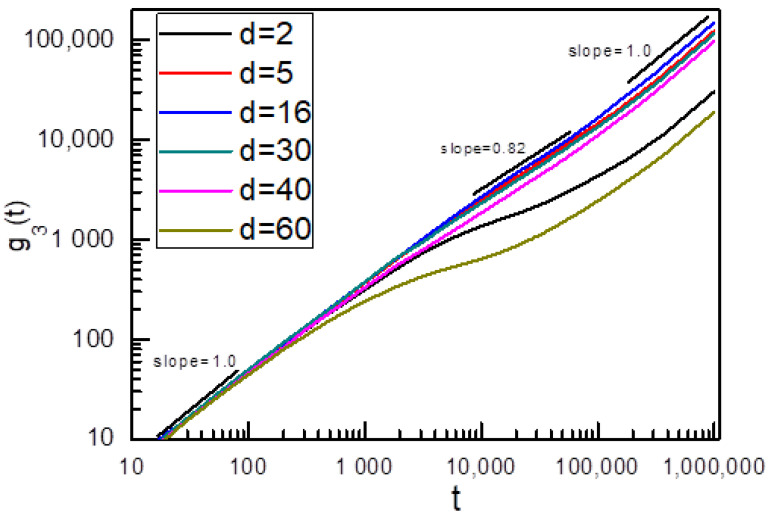
Time dependence ofg_3_(t) of ring chainssliding along the symmetric fluctuating axial chains (K_1_ = K_2_ = 0) with different distances d between two symmetric fluctuating axialchains.

**Figure 5 polymers-14-00079-f005:**
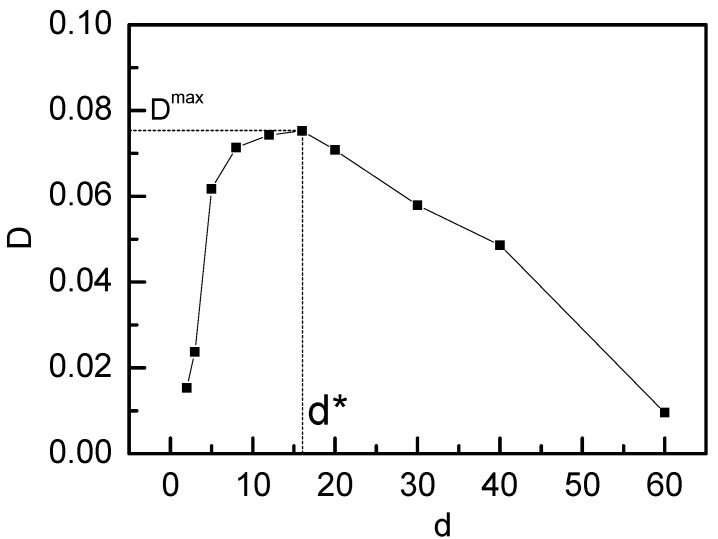
Diffusion coefficient Dof ring chain on the symmetric fluctuating axial chains (K_1_
*=* K_2_
*=* 0) with different distances d between two symmetric fluctuating axial chains.

**Figure 6 polymers-14-00079-f006:**
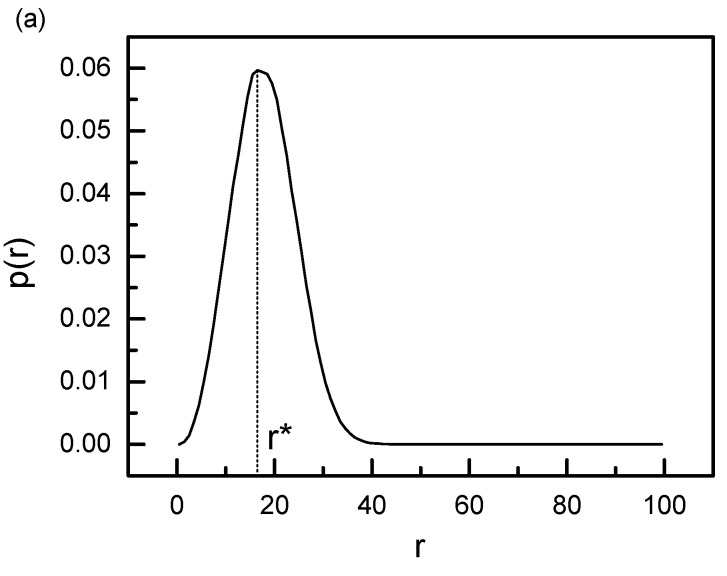
(**a**) Probability distribution function P(r) of end-to-end distance r for a single 100-monomer flexible polymer chain. Here the maximum value of P(r) is located at r* = 16.5. (**b**) A schematic illustration of two ring chains sliding along the symmetric axial chains fastest at d = d* = r* owing to the fact that the conformational entropy of the linking chain between two ring chains approaches is close to the maximum value, and the entropy force approaches to the minimum one at d = d*.

**Figure 7 polymers-14-00079-f007:**
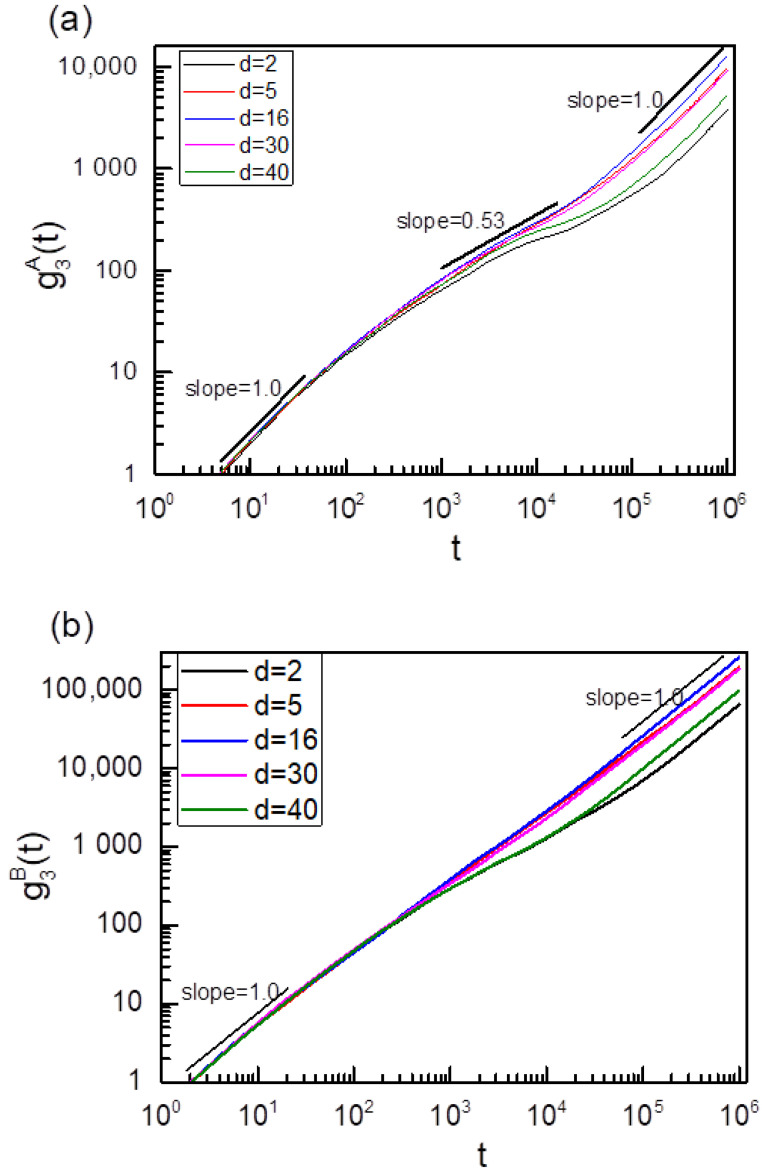
(**a**) Time dependence of $g_{3}^{A}\left(t\right)$ of ring chain Asliding alongthe fixed rod-like chains (K_1_ = ∞), and (**b**) time dependence of $g_{3}^{B}\left(t\right)$ of ring chain Bsliding along the fluctuating chain (K_2_ = 0) for two ring chains on the asymmetric axial chains with different distances d between two asymmetric axial chains.

**Figure 8 polymers-14-00079-f008:**
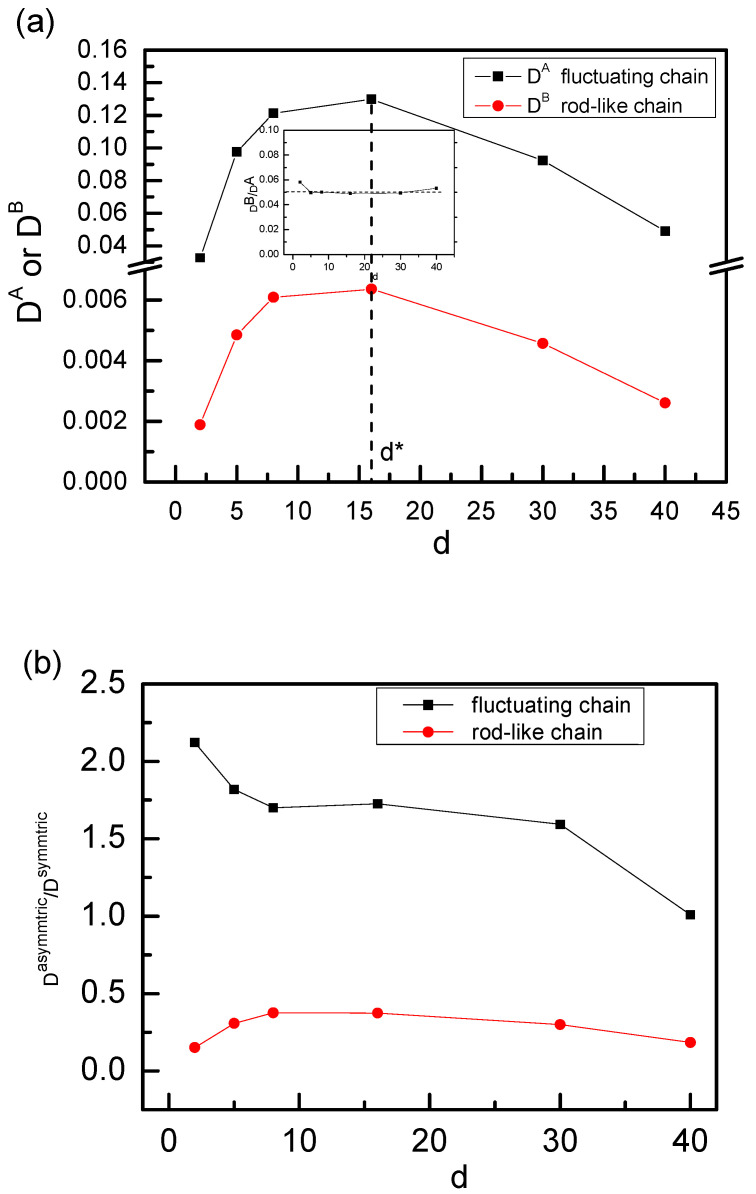
(**a**) Diffusion coefficient Dof ring chainsalongthe asymmetricaxial chainsfor ring chain A on fixed rod-like axial chain D^A^ and ring chain B on fluctuating axial chain D^B^ with different distances d between two asymmetricaxial chains. The inset figure shows the ratio of D^B^/D^A^ as a function of d. (**b**) Ratio of D^asymmtric^/D^symmtric^ for ring chain alongfluctuating axial chain or ring chain along the fixed rod-like chain in the asymmetric and symmetric axial chains cases with different distances d between two axial chains.

**Figure 9 polymers-14-00079-f009:**
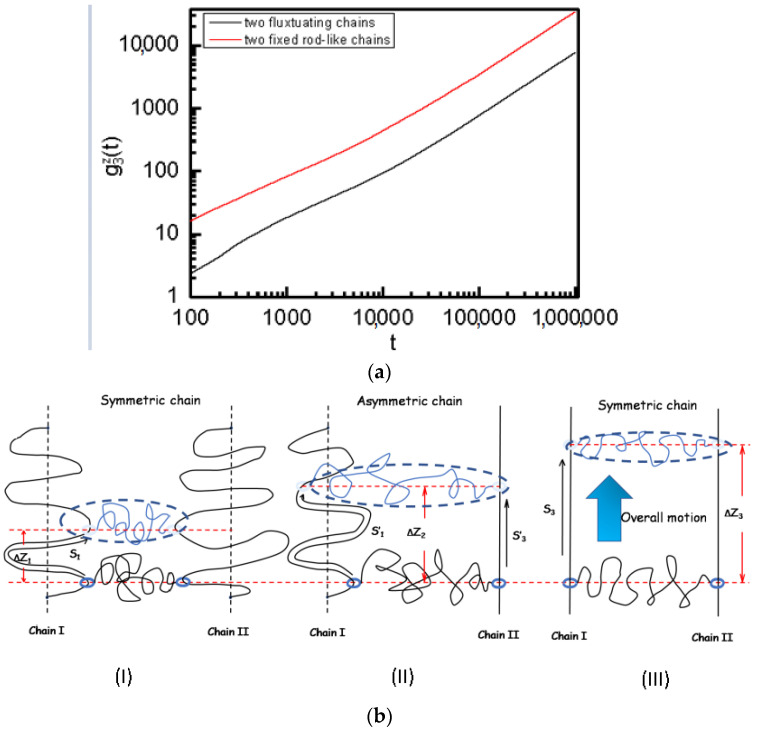
(**a**) The absolute mean-square displacement (MSD), $g_{3}^{z}\left(t\right)$, of the ring from the center of mass of the ring in the *z*-axis direction for ring chain along two symmetric fluctuating axial chains and two symmetric fixed rod-like axial chains. The definition of the *z*-axis direction is given in Figure 1. (**b**) A schematic illustration to explain why ring chain slide along fixed rod-like axial chain on the asymmetric chain slower than that on the symmetric chain (i.e., S′_3_ < S_3_) and why ring chain slide along fluctuating axial chain on the asymmetric chain faster than that on the symmetric chain (S′_1_ > S_1_).

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
