# Peer review of "Sliding Dynamics of Ring Chains on Two Asymmetric/Symmetric Chains in a Simple Slide-Ring Gel"

_polymers, 2021, doi:10.3390/polym14010079_

Round 1
Reviewer 1 Report
It is a nice computational study of rings dynamics performed by the molecular-dynamics coarse-grained modelling. LAMMPS software was, used and well-known Kremer-Grest models have been explored. The manuscript contains new and interesting results, I recommend the publication. I have only minor suggestions, the additional review, from my point of view, is not necessary.
- The Abstract is way too long, and could be shortened.
- in Figure 1 it would be nice to show the real snapshot(s) of the simuklated systems, not only the schematic diagram (this diagram is not the Kremer-Grest model, and the Figure is used mainly to introduce this model...)
- page 3: K1 and K2 are not bending energies, they are bending constants. Their values should be given.
- page 4, above Eq. 6. It is not clear what do the words "in the coordinate index along the chain" mean? The distance measured in these units, or something else? Please clarify.
- How sensitive are the presented results to the specific choice of the parameters (the chains lengths , the bending constants)?
Reviewer 2 Report
See attached pdf file.

Round 2
Reviewer 2 Report
The authors have significantly improved their manuscript and I now recommend its publication in Polymers in the present form.